# Comprehensive Characterization of Human Lung Large Cell Carcinoma Identifies Transcriptomic Signatures with Potential Implications in Response to Immunotherapy

**DOI:** 10.3390/jcm11061500

**Published:** 2022-03-09

**Authors:** Javier Ramos-Paradas, David Gómez-Sánchez, Aranzazu Rosado, Alvaro C. Ucero, Irene Ferrer, Ricardo García-Luján, Jon Zugazagoitia, Nuria Carrizo, Ana B. Enguita, Esther Conde, Eva M. Garrido-Martin, Luis Paz-Ares

**Affiliations:** 1H12O-CNIO Lung Cancer Clinical Research Unit, Health Research Institute Hospital 12 de Octubre (imas12)/Spanish National Cancer Research Center (CNIO), 28041 Madrid, Spain; ramosparadasjavier@gmail.com (J.R.-P.); dagsbio@gmail.com (D.G.-S.); arosadodiez@gmail.com (A.R.); acuceroh@ucm.es (A.C.U.); ireneferrersan@gmail.com (I.F.); nuria.carrizo.mijarra@gmail.com (N.C.); lpazaresr@seom.org (L.P.-A.); 2Spanish Center for Biomedical Research Network in Oncology (CIBERONC), 28029 Madrid, Spain; esthercondegallego@gmail.com; 3Faculty of Medicine, Complutense University, 28040 Madrid, Spain; 4Pulmonary Department, 12 de Octubre Hospital, 28041 Madrid, Spain; rglujan@hotmail.com; 5Medical Oncology Department, 12 de Octubre Hospital, 28041 Madrid, Spain; jonzuga@gmail.com; 6Pathology Department, 12 de Octubre Hospital, 28041 Madrid, Spain; abenguita@hotmail.com

**Keywords:** non-small cell lung cancer, lung large cell carcinoma, immunotherapy, checkpoint inhibitors, multiparametric analysis, predictive biomarkers

## Abstract

Lung cancer is the leading cause of cancer mortality worldwide, with non-small cell lung cancer (NSCLC) being the most prevalent histology. While immunotherapy with checkpoint inhibitors has shown outstanding results in NSCLC, the precise identification of responders remains a major challenge. Most studies attempting to overcome this handicap have focused on adenocarcinomas or squamous cell carcinomas. Among NSCLC subtypes, the molecular and immune characteristics of lung large cell carcinoma (LCC), which represents 10% of NSCLC cases, are not well defined. We hypothesized that specific molecular aberrations may impact the immune microenvironment in LCC and, consequently, the response to immunotherapy. To that end, it is particularly relevant to thoroughly describe the molecular genotype–immunophenotype association in LCC–to identify robust predictive biomarkers and improve potential benefits from immunotherapy. We established a cohort of 18 early-stage, clinically annotated, LCC cases. Their molecular and immune features were comprehensively characterized by genomic and immune-targeted sequencing panels along with immunohistochemistry of immune cell populations. Unbiased clustering defined two novel subgroups of LCC. Pro-immunogenic tumors accumulated certain molecular alterations, showed higher immune infiltration and upregulated genes involved in potentiating immune responses when compared to pro-tumorigenic samples, which favored tumoral progression. This classification identified a set of biomarkers that could potentially predict response to immunotherapy. These results could improve patient selection and expand potential benefits from immunotherapy.

## 1. Introduction

Lung cancer is responsible for the second highest incidence of tumors worldwide, representing 11.4% of newly diagnosed cancer cases in 2020. It is also the leading cause of cancer-related deaths, accounting for 20% of the total [1]. The histology of lung cancers is divided into two main categories: small cell lung cancer (SCLC), which represents 15% of all lung cancer cases, and non-small cell lung cancer (NSCLC) which accounts for the remaining 85% of cases. This NSCLC subtype can subsequently be subdivided into the categories of adenocarcinoma (ADC), squamous cell carcinoma (SCC) and large cell carcinoma (LCC) [2], the latter of which is conventionally considered to represent around 10% of all NSCLC cases. Lung LCC is defined as a lung tumor that does not present characteristics of ADC, SCC or SCLC [3]. However, this vague definition encompasses a broad and heterogeneous set of lung tumors, thereby complicating patient classification [4,5] and limiting scientific advances in the development of diagnostic and treatment strategies.

Currently, LCC diagnosis and classification relies on the evaluation of tissue morphology, which requires a resected tumor [3], and the expression of ADC, SCC and SCLC markers, as well as molecular signatures, which are often limited by tissue availability [4,5,6,7]. Recent research has suggested a new subclassification of LCC into LCC-ADC (positive for ADC markers), LCC-SCC (positive for SCC markers) and LCC-Null (negative for both) [8,9].

However, a thorough description of LCC is lacking. Several reports have evaluated the molecular landscape of LCC and concluded that this histology presents specific molecular features with potential implications in survival and treatment decisions [8,10,11,12,13]. Other studies have identified gene expression profiles associated with specific NSCLC histologies [14,15,16,17]. One article proposed that the phenotypic and histological features of lung cancer are associated with these transcriptional signatures rather than its genomic characteristics [18], suggesting the importance of evaluating the expression profile of lung tumors. In the context of LCC, some genes have shown specific levels of expression in LCC when compared to adenocarcinomas or squamous cell carcinomas [19]. One report demonstrated that 90% of LCCs could be properly classified according to a distinct expression profile, which was also associated with specific molecular and clinical features [20]. However, these studies described a very reduced number of LCC tumors and their findings require validation in larger cohorts. Furthermore, most of these multifactorial studies mainly describe adenocarcinomas and squamous cell carcinomas. Therefore, there is an urgent need to perform more comprehensive studies focused on LCC tumors.

While treatment options in LCC are, in general terms, similar to those of other NSCLC histologies, including surgical resection in the early stages, chemotherapy, radiotherapy, etc., no targeted therapies are currently available for these patients. LCC tumors can harbor actionable mutations such as those affecting the epidermal growth factor receptor (EGFR), which would offer the possibility of employing personalized medicine for these patients [8]. In recent years, immunotherapy with checkpoint inhibitors has shown outstanding results in advanced NSCLC. Programmed cell death protein 1/programmed cell death ligand 1 (PD-1/PD-L1) inhibitors, administered both in monotherapy or in combination with chemotherapy, and cytotoxic T-lymphocyte associated protein 4 (CTLA-4) inhibitors have shown a consistent benefit in pre-treated and untreated patients [21,22,23,24,25,26], thus shaping new directions in the treatment of NSCLC, particularly for the ADC and SCC subtypes. Further to this, the currently approved first line of treatment for NSCLC is immunotherapy alone for those patients with a high PD-L1 expression (≥50%) and a combination of chemotherapy + immunotherapy if the PD-L1 expression is below 50% [24,27,28,29], as long as no actionable molecular alterations can be targeted [30]. These treatment guidelines are also applied to LCC patients.

However, the identification of likely responders to immunotherapy remains a major research challenge in oncology, given that the use of PD-L1 expression in tumor cells as the only approved biomarker in NSCLC presents several limitations [21,31]. First, PD-L1 expression is heterogeneous since it can be present in different cell populations and cell compartments. Second, PD-L1 expression can be altered by previous treatment lines, tumor heterogeneity and other factors. And third, despite countless efforts, the protocol for PD-L1 evaluation is yet to be standardized across clinics [32,33]. Overall, PD-L1 expression is neither specific nor sensitive enough to predict patient response to immunotherapy [21], thus making the identification of new biomarkers an unmet need, urgently requiring attention.

One notable example of a potential biomarker for NSCLC is the tumor mutational burden (TMB), which is defined as the total number of mutations per megabase of the tumor genome-encoding area [34,35,36]. TMB has shown promise as a predictive biomarker in NSCLC, a tumor type featuring a high prevalence of somatic mutations [37,38,39]. Theoretically, a higher number of mutations would increase the presence of tumor neoantigens that would trigger the immune response [35]. However, TMB presents with similar pitfalls to PD-L1 expression as a predictor of response to immunotherapy [21,31]. Other examples of immunotherapy response biomarkers under investigation include the density, tissue location and functional phenotype of tumor infiltrating lymphocytes (TILs) [31,40], HLA-I diversity [31,41], transcriptional and epigenetic signatures and even the microbiome [31]. Importantly, many other components of the immune system beyond lymphocytes could influence responses to checkpoint inhibitors [31,42]. As cancer complexity is likely to be the reason why a single optimal biomarker to predict response to immunotherapy remains elusive, many studies have therefore focused on searching for a multiparametric biomarker signature, based on clinical, molecular and immune characteristics.

In general terms, the LCC histology is exhibited by a subgroup of patients with poor prognosis that encompass a heterogeneous set of tumors lacking a proper classification. In the era of personalized medicine, the application of tailored treatments according to tumor histology and its molecular/immune landscape, employing targeted therapies or immunotherapies requires a more exhaustive description of LCC [6]. We hypothesized that there may be an association between the underlying molecular aberrations of LCC tumor cells and the immune phenotype in the tumor microenvironment. The aim of this study was thus to confirm such an association and to explore its implications in the treatment decision. From a cohort of N = 200 patients with NSCLC tumors, we gathered tumor samples from 18 of these patients with a diagnosis of early-stage LCC, collected in our hospital from 2002 to 2011. Most of the cohort was treatment naïve at the time of the surgical resection. This clinically annotated cohort was subjected to a comprehensive clinical, molecular and immune evaluation. Associated parameters arising from genomic, transcriptomic and immunophenotypic results allowed the identification of subgroups of patients characterized by a specific set of molecular and immune features.

This work provides an important step towards a comprehensive description of LCC as a proper entity with specific underlying features. The identification of patient subsets could enhance treatment decision outcomes and ultimately improve the benefits obtained from immunotherapy.

## 2. Materials and Methods

### 2.1. Lung Large Cell Carcinoma Samples and Patient Cohort

We established a cohort of 18 patients from whom early-stage lung LCC tumors were resected. After signed consent had been provided by the patients, resected tumor samples were formalin-fixed and paraffin-embedded (FFPE) and fresh-frozen with optimal cutting temperature (OCT) compound and stored in the Pathology Department of the 12 de Octubre University Hospital. The described procedures were approved by the Ethics Committee of the 12 de Octubre Hospital with Identification Number #17/454. All tumor samples were successfully described with all the applied methodologies, a summary of which can be found in Figure 1. Clinical and demographic information of the patient cohort is presented in Table 1. Samples were required to fulfill the following criteria in order to be eligible for the study: LCC histology, early stage, sufficient sample availability, complete clinical annotation and agreed consent provided by the patient for further analyses to be performed.

### 2.2. DNA Extraction, Quantification and Quality Measurement

For each of the 18 tumor samples from the cohort, 5 sections (thickness 4 μm) of fresh-frozen tissue in OCT (−80 °C) were used to extract DNA. For this purpose, the DNeasy Blood and Tissue kit (#69504, QIAGEN, Hilden, Germany) was used according to the manufacturer’s instructions. After DNA extraction, quantification was performed with the Quantifluor dsDNA system (#E2670, Promega, Madrid, Spain), with an average yield of 75.07 ng/μL obtained for the 18 samples, surpassing quantity requirements for subsequent applications. Finally, DNA quality was determined with the Illumina FFPE QC kit (#15013664, Illumina, San Diego, CA, USA), comparing the amplification efficiency to a control DNA provided by the manufacturer. IQ SYBR Green Supermix (#170-8880, Biorad, Hercules, CA, USA) was used to perform the amplification reaction. All DNA samples showed adequate quality required for use in subsequent protocols, and all procedures were performed by the same person at the 12 de Octubre Hospital to ensure experimental consistency. Information related to DNA extraction, quantification and quality can be found in Appendix A.

### 2.3. RNA Extraction, Quantification and Quality Measurement

For the 18 tumor samples from the cohort, a further 5 sections (thickness 4 μm) of fresh-frozen tissue in OCT (−80 °C) were used to extract RNA. To accomplish this goal, the RNeasy Mini kit (#74104, QIAGEN, Hilden, Germany) was used, according to the manufacturer’s instructions. RNA quantification was performed with the Quantifluor RNA system (#E3310, Promega, Madrid, Spain), with an average yield of 141.09 ng/μL obtained, again exceeding the amounts required for subsequent procedures. RNA quality was evaluated using the RNA 6000 Nano kit (#5067-1511, Agilent, Santa Clara, CA, USA) or RNA 6000 Pico kit (#5067-1513, Agilent, Santa Clara, CA, USA) depending on the RNA concentration. According to their RNA Integrity Number (RIN), all samples were of sufficient quality for use in subsequent methodologies. All procedures were performed by the same person at the 12 de Octubre Hospital as described above. Information related to RNA extraction, quantification and quality can be found in Appendix A.

### 2.4. Tumor Genetic Sequencing

Specific molecular aberrations in the 18 samples were detected with the TruSight Tumor 170 kit (#20028821, Illumina, San Diego, CA, USA), a panel that targets 170 genes frequently mutated in cancer. For this purpose, both DNA (to detect single nucleotide variants (SNVs), copy number variants (CNVs) and indels) and RNA (to detect fusions and splice variants) were used. The DNA input was 120 ng and the RNA input was 85 ng per sample. Molecular aberrations in all samples were successfully described. Libraries were prepared according to the manufacturer’s instructions. Although initial steps for DNA and RNA sample handling were different, subsequent steps were identical for both types of input.

Briefly, DNA was sheared with an M-220 Ultrasonicator (Covaris). Total DNA input was 120 ng in a total volume of 52 μL (DNA fragmentation concentration = 2.31 ng/μL). Samples were sheared in microTUBEs–50 AFA Fiber Screw Cap (#520166, Covaris, Woburn, MA, USA) and AFA–Grade Water (#52101, Covaris, Woburn, MA, USA). An M220 Holder XTU (PN 500414) and an M220 holder XTU Insert microtube 50 µL (PN 500488) were used. With respect to the experimental settings, a peak power of 75 W was used, with a duty factor of 15% and 1000 cycles per burst for a total treatment of 360 s per sample at a temperature of 20 °C. Sheared DNA fragments were analyzed with a High Sensitivity DNA kit (#5067-4626, Agilent, Santa Clara, CA, USA) and a Bioanalyzer system (Agilent), to evaluate the peak size of the generated fragments.

RNA samples were retrotranscribed as a first step with a total input of 85 ng per sample, following the manufacturer’s instructions.

From this point, both DNA and RNA samples were treated in the same manner. Briefly, an end repair and an A-tailing step to convert the 5′ and 3′ overhangs into blunt ends were performed. The 3′ to 5′ exonuclease activity removed the 3′ overhangs and the 5′ to 3′ polymerase activity filled in the 5′ overhangs. The 3′ ends became A-tailed during this reaction in order to prevent undesired ligations. The ligation of adapters was then performed, followed by the removal of the remaining ligation reagents. Afterwards, indices to identify each library were added to every sample. Once libraries were generated, enrichment steps were performed. A hybridization-capture was performed (twice), in which a pool of oligos specific to 170 genes was hybridized to the generated libraries, which were then captured with biotin probes attached to streptavidin magnetic beads. Once the regions of interest were enriched, amplification procedures were performed. Thereafter, amplified libraries were quantified with the Qubit dsDNA HS Assay kit (#Q32854, Invitrogen, Waltham, MA, USA). A library concentration of at least 3 ng/μL was necessary to ensure an efficient bead-based library normalization. Quality control and library preparation details are provided in Appendix A.

Sequencing was carried out in a NextSeq 500 sequencer with NextSeq 500/550 High Output kit v2.5 reagents, using 300 cycles (High Output kit #20024908, Illumina, San Diego, CA, USA). PhiX control V3 (#150117666, Illumina, San Diego, CA, USA) was used as a sequencing control. An initial data analysis was performed with the TruSight Tumor 170 application in BaseSpace (Illumina). Sample sequencing metrics can be found in Appendix A. Molecular alterations detected for each sample are described in Appendix A.

### 2.5. Gene Expression Analysis with the Oncomine Immune Response Research Assay

The expression of 395 genes involved in the tumor–immune system communication was evaluated with the Oncomine Immune Response Research Assay (OIRRA) (#A32928, ThermoFisher Scientific, Waltham, MA, USA). An input of 10 ng of RNA was used for each sample. Briefly, a retro-transcription step was performed with the SuperScript Vilo cDNA Synthesis kit (#11754-250, Invitrogen, Waltham, MA, USA). Libraries were then generated with the automated Ion Chef System (ThermoFisher Scientific) with the following settings: number of groups of primers = 1, amplification cycles of target genes = 17, annealing and extension time = 4 min. Resulting libraries (8 libraries per run) were quantified with the Ion Library TaqMan Quantitation kit (#4468802, Applied Biosystems, Waltham, MA, USA) and diluted to a concentration of 33 pM in a total volume of 25 μL, thereby assuring maximum monoclonality with the used sequencing chip (Ion 530^TM^ Chip, ThermoFisher Scientific, Waltham, MA, USA). Automated library template and chip loading was subsequently performed with the Ion Chef System. Finally, libraries were sequenced in an Ion S5 System (ThermoFisher Scientific). Among the resulting files, absolute read counts were used to perform subsequent analyses, as described below. Quality control and library preparation details are shown in Appendix A. Raw and normalized gene expression data can be found in Appendix A.

### 2.6. Immunohistochemistry

FFPE tumor samples were sectioned at a thickness of 3.5 μm and mounted on BOND Plus Slides (#S21.2112.A, Leica, Wetzlar, Germany). The protocol consisted of epitope retrieval and staining (antibody incubation, peroxide block, post primary incubation, mixed Diaminobenzidine (DAB) and hematoxylin) and was carried out in a Bond-III IHQ-ISH Automated Station (Leica). Primary antibodies used were: CD20 (DAKO, M0755, 1/50), CD4 (Leica, NCL-L-CD4-368, 1/50), CD68 (DAKO, M0814, 1/1000), CD8 (DAKO, M7103, 1/19), TTF 1 (Leica, TTF-1-L-CE, 1/100) and p40 (PA0163, pre-diluted, Leica, Wetzlar, Germany). Primary antibodies were diluted in Primary Antibody Diluent (#AR9352, Leica, Wetzlar, Germany). For antigen retrieval, Epitope Retrieval Solution 1 (ER1, #AR9961, Leica, Wetzlar, Germany) and Epitope Retrieval Solution 2 (ER2, #AR9640, Leica, Wetzlar, Germany) were used. Specific retrieval conditions for each marker were ER1 buffer (pH6) for CD20, CD68 and CD8 and ER2 buffer (pH9) for CD4, TTF-1 and p40. Retrieval times were 20 min for CD20 and CD4, 10 min for CD68 and TTF-1, and 30 min for CD8 and p40. The incubation time was 20 min for all antibodies. Visualization was performed with the BOND Polymer Refine Detection (#DS9800-CN, Leica, Wetzlar, Germany), which uses DAB as the chromogen to detect the marker and hematoxylin as a counterstain. Positive staining for each marker was confirmed in parallel using human tonsil tissue.

Immunohistochemistry scores from all markers analyzed can be found in Appendix A. Three comparison groups of samples were selected based on the expression of these markers according to their assigned score: Score 0 (<1% of expression), Score 1 (1–10% of expression) and Score 2 (>10% of expression).

In the case of staining for PD-L1, FFPE sections were stained with anti-PD-L1 22C3 mouse monoclonal primary antibody by utilizing the EnVision FLEX visualization system on a Dako Autostainer Link 48 system with negative control reagents and cell line run controls as described in the PD-L1 IHC 22C3 pharmDx package insert [43].

### 2.7. Bioinformatic Analyses

For the analysis of molecular data, custom scripts were used for annotation and filtering of small variants (SNVs and indels). The following public databases were used for interpretation: gnomAD, Cosmic, ClinVar and an in-house database of allele frequencies from non-cancer patients. We filtered out variants with a minor allele frequency (MAF) ≥0.1% and non-exonic variants. Finally, the Integrative Genomics Viewer (IGV) was used for manual review of the remaining set of variants. CNVs, fusions and splice variants were considered as valid only when reported as high-confident by the TruSight Tumor 170 application (Illumina). Frequently altered genes described in the literature and the most frequently mutated genes in our cohort were represented in the figures shown.

Raw transcriptomic data from the OIRRA (absolute read counts) were normalized with the Variance Stabilizing Transformation (VST) algorithm, using the DESeq2 R package [44]. Following data normalization, tumor clusters and gene clusters were defined with the algorithm ‘Consensus Clustering’ using the ConsensusClusterPlus R package, through 10,000 and 1000 iterations, respectively [45].

The over-representation analysis (ORA) was performed based on gene categories defined by ThermoFisher for the OIRRA. Briefly, a comparison between the obtained percentage of genes of a certain category versus the expected percentage was performed. ORA pie charts were constructed using the Caroline R package [46]. Statistical analyses of over/under-representation were performed with a Fisher’s exact test.

Differential gene expression analysis between the two LCC clusters of tumors was evaluated using the limma R package and its “voom” function specifically designed for RNA-Seq data. Genes with a false discovery rate (FDR) ≤ 0.05 and a Log_2_ Fold Change ≥ |1| were considered as significant and used in subsequent analyses.

Since the number of significant molecules was extensive, we selected our set of potential biomarkers based on the following criteria: (1) Strict statistical significance: only genes differentially expressed (FDR ≤ 0.01 and a log_2_ Fold Change ≥ |1|) were considered; (2) Biological role related to immunomodulation or tumor progression; (3) Genes suggested as predictive of response to immunotherapy as described in previous literature.

### 2.8. Statistical Analyses

Statistical analyses were performed with GraphPad Prism. Chi Square/Fisher’s exact test were used for categorical data and log-rank test was used for survival analyses. Multivariant analyses were performed with SPSS using a Chi Square test.

### 2.9. Experimental Design

The objective of the study was to perform a complete description of the molecular and immune profiles of resected samples (Figure 1). The detailed molecular characteristics of the cohort were described with a targeted DNA-Seq panel (TruSight Tumor 170, TST170) that detects molecular aberrations in 170 genes, including SNVs, CNVs, insertions, deletions, amplifications, rearrangements, fusions and splice variants. The immune features of the cohort were evaluated with a targeted RNA-Seq panel which evaluated the expression of 395 genes involved in the tumor–immune system communication (OIRRA). In parallel, we evaluated the tumor immune infiltrate level of B cells, T cells and macrophages, and the expression of the immunomodulatory protein PD-L1 by immunohistochemistry, which is the only currently approved predictive biomarker for response to immunotherapy in NSCLC. This multiparametric clinical, molecular and immune data were evaluated with integrative bioinformatic pipelines in order to define novel subgroups of LCC and to identify new candidate predictive biomarkers for response to immunotherapy (Figure 1).

## 3. Results

### 3.1. Description of the LCC Cohort

Demographic and clinical characteristics of the patient cohort can be found in Table 1. Briefly, most of the LCC patients in the cohort were males (16/18) and diagnosed with chronic obstructive pulmonary disease (COPD, 10/18). All the LCC tumors were early-stage and half of them were stage II. Interestingly, all patients were current or former smokers, with no never-smokers in the cohort. Some tumors had molecular aberrations in frequently altered genes in NSCLC such as *ALK*, *EGFR*, *KRAS* or *ROS1*, as determined with the TruSight Tumor 170 sequencing panel in-house. However, it is important to emphasize that all these genetic alterations were of uncertain significance in our bioinformatic analysis with the Varsome database, with the exception of a G12C mutation in *KRAS*, which may lead to treatment adaptation, considering the new *KRAS* inhibitors under investigation [47] and the potential utility of targeted therapies in LCC as proposed by previous research [8]. These specific molecular alterations can be found in Appendix A. Most patients (15/18) were treatment-naïve at the time of surgery, with only one case confirmed for previous treatment. Half of the patients relapsed and the majority of the patients died by the end of the study period (Figure 1, Table 1).

### 3.2. Evaluation of the Level of Infiltration of Immune Cell Populations and the Expression of PD-L1 Detected by Immunohistochemistry

To evaluate the infiltration of various immune cell populations and to identify expression of the immunomodulatory protein PD-L1 we stained five slides from each tumor sample with the markers for the following: CD4^+^ T cells, CD8^+^ T cells, CD20^+^ B cells, CD68^+^ macrophage/monocytic populations and PD-L1. Each immune cell marker was evaluated with the following score: 0 (<1% of expression, negative), 1 (1–10%, low expression) and 2 (>10%, high expression) (Figure 2A). PD-L1 was evaluated as negative (<1%) or positive (≥1%), with the percentage of expression specified (Figure 2B).

Taking the whole cohort (N = 18) into consideration, the most abundant infiltrating immune cells were macrophages (100% of tumors with a score of 1 or above), followed by the T CD8^+^ cells (83% of tumors with a score of 1 or more) and B cells (78% of tumors with a score of 1 or more). T CD4^+^ cells were the least infiltrating immune cell population, with only a 22% of tumors with a score of at least 1. Only a small fraction of patients (17%) was positive for PD-L1 expression based on a positivity cutoff of 1% (Appendix A).

We also examined the possible correlation between the PD-L1 expression and the level of infiltration of the evaluated immune cell populations. The onlythree patients who were positive for PD-L1 expression were also positive for T CD8^+^ infiltration and negative for T CD4^+^ infiltration, which suggests a connection between an ongoing immune response with the tumor cell immune evasion mechanisms. Nonetheless, these observations need to be confirmed in larger cohorts of LCC with more PD-L1^+^ tumors. No clear relationship was found between the PD-L1 expression and the infiltration of B cells (CD20^+^) or macrophages (CD68^+^). These results can be found in Appendix A.

The whole cohort was classified as LCC according to the 2004 WHO guidelines, since all tumors lacked morphological differentiation. Of the eighteen cases, nine did not express any adenocarcinoma or squamous cell carcinoma markers and were therefore re-classified as LCC-Null, following the 2015 WHO guidelines [3]. Four tumors were positive for TTF-1 expression and were re-classified as adenocarcinomas. Five tumors were positive for p40 and were re-classified as squamous cell carcinomas. However, the frontiers of the LCC histology are yet to be well established, beyond a pathological perspective. Therefore, we followed the approach described in a recently published report [8] and reclassified these tumors as LCC-ADC or LCC-SCC, respectively. TTF-1 and p40 results can be found in Appendix A.

### 3.3. Definition of Novel LCC Tumor Subgroups by Consensus Clustering with Specific Molecular, Immune and Clinical Features

We employed a publicly available algorithm named ‘Consensus Clustering’ that allowed the clustering of patients and genes based on expression data obtained from the OIRRA assay, i.e., according to tumor–immune system crosstalk. Briefly, this method performs a hierarchical clustering that distributes the information optimally. Importantly, the output classification is highly reproducible, contrary to most methods based on hierarchical clustering. As a result, we defined two novel LCC tumor subgroups and three clusters of genes (Figure 3A). These groups were robustly obtained through 10,000 iterations for patients and 1000 iterations for genes. By assessing the functional characteristics of the three different gene clusters, we identified a transcriptional orientation of the patient subgroups to be either more immunogenic (named ‘pro-immunogenic tumors’) or less immunogenic (named ‘pro-tumorigenic tumors’) (Figure 3A). This was addressed through an ORA (Figure 3B), in which gene clusters A and C showed an over-representation of gene functions related to immune processes, such as lymphocyte infiltration, type II interferon signaling, antigen processing, TCR co-expression, cytokine signaling, B cell markers and innate immune responses. Gene cluster B, on the other hand, showed over-represented genes related to tumor markers and antigens, adhesion and migration (Figure 3B). The functional categories that reached a statistically significant over or under-representation are shown in Figure 3B. The pro-tumorigenic group was characterized by an upregulation of gene cluster B and a downregulation of gene cluster A, the opposite to that of the pro-immunogenic group. The gene expression profile of cluster C was not different between groups. Representative genes of each cluster and selected OIRRA gene categories can be found in Appendix A. A complete list of the OIRRA genes, their cluster and their gene category can be found in Appendix A.

We compared the clinical, molecular and immune annotations between the pro-immunogenic and the pro-tumorigenic subgroups (Figure 3A), but none of these parameters were significantly different between the two groups, which is likely due to the limited number of patients in the study. However, certain annotations suggested that molecular and immune events were occurring predominantly in one subgroup or the other. While the pro-tumorigenic group accumulated patients with alterations in *RET*, *PIK3CA*, *PIK3CB* and *FGF10* and a higher proportion of relapsed patients, the pro-immunogenic group encompassed tumors with an alteration in *ALK*, *ARID1A*, *MET*, *KRAS* or *FGF3* and with a high level of T CD4^+^ or B cell infiltration (above 10%) (Figure 3A). In addition, the pro-immunogenic group included more patients in which all of immune cell populations evaluated were present, which was more apparent for CD8^+^ T cells and CD20^+^ B cells (Figure 3C and Table 2). Importantly, no bias was observed with respect to the subtypes of LCC (LCC-ADC, LCC-SCC and LCC-Null) since they were not significantly associated with any specific subgroup and showed an even distribution.

### 3.4. Analysis of Differentially Expressed Genes between the Pro-Immunogenic and Pro-Tumorigenic LCC Subgroups

Once tumors had been classified as pro-immunogenic or pro-tumorigenic based on the transcriptomic profile of the tumor–immune system crosstalk, we performed differential gene expression analyses to identify which genes characterized each subgroup according to their significant up- or down-regulation. For this purpose, we applied the Limma-Voom algorithm and identified 152 genes with significantly different expression between the pro-immunogenic and the pro-tumorigenic subgroups based on an FDR ≤ 0.05 and a log_2_ Fold Change ≥ |1| as statistical cut-offs. Of these 152 genes, 115 were significantly upregulated in the pro-immunogenic tumors, while 37 were significantly upregulated in the pro-tumorigenic tumors (Figure 4A).

Consistent with previous analyses, most upregulated genes in pro-immunogenic tumors were immune mediators involved in functions such as antigen processing and presentation (*CD1C*, *HLA-C* and other HLA genes), B cell markers and signaling (*CD19*, *MS4A1*, *FCRLA*), chemokine signaling (*CCL20*, *CXCR5*), cytokine signaling (*IL1A*, *IL18*), infiltration and activation of lymphocytes (*ICOS*, *CD53*, *TAGAP*), TCR signaling (*ZAP70*) and TCR co-expression (*CD247*, *CD3D*), among others (Figure 4B,D). Pro-tumorigenic tumors displayed upregulation of genes related to pro-tumoral processes such as adhesion or migration (*NCAM1*), proliferation (*CDK1*, *TOP2A*) or extensively described tumor markers (*CDKN2A*, *MTOR*), among others, while showing an impairment in immune-related functions (Figure 4C,E). Data for the full differential expression analysis can be found in Appendix A.

### 3.5. Clinical Relevance of the Defined LCC Subgroups

We addressed whether the transcriptomic profile of the defined LCC subgroups was relevant for the clinical outcome regardless of the treatment received. Therefore, we sought to identify any possible associations between the tumor subgroups and progression-free survival (PFS) (Figure 5A) and overall survival (OS) (Figure 5B) data, but no significant differences were observed. However, despite the reduced number of patients, the analysis suggested a clear trend towards a greater PFS and OS in the pro-immunogenic group, as might be expected.

A PFS analysis according to the stage of the tumors and their transcriptomic subgroup suggested that these factors had an impact on relapse. Pro-tumorigenic tumors with stage III were those with a lower PFS (Appendix A).

### 3.6. Transcriptional Signatures of the Defined LCC Subgroups and Potential Implications in Response to Immunotherapy

We subsequently identified a transcriptomic signature with a reduced number of genes that could be used to classify patients in the defined pro-immunogenic and pro-tumorigenic subgroups in order to translate our findings into an experimentally applicable test with possible clinical interest. This signature consists of 20 differentially expressed genes between the subgroups with known roles in immune response and tumor progression (Figure 5C,D). Genes in this selection that are distinctive of pro-immunogenic tumors were involved in adaptive immune response stimulation, apoptosis, immune chemotaxis, immune response regulation and iron metabolism. Conversely, genes characteristic of the pro-tumorigenic tumors were related to cell cycle regulation, DNA replication and repair and iron metabolism. In summary, genes related to pro-immunogenic tumors favored immune functions, while those related to pro-tumorigenic tumors supported tumoral progression (Figure 5C,D).

The final objective of this study was to identify predictive biomarkers of response to immunotherapy in LCC. Ideally, the identification of these biomarkers should be performed in an advanced stage LCC cohort treated with immunotherapy. However, this approach presents two main handicaps in the current lung cancer clinical practice. First, the number of advanced LCC patients undergoing immunotherapy treatment is quite reduced. Second, for most of these cases the amount and quality of available tumor tissue would not allow a comprehensive description of LCC such as the one provided in this work. In this context, findings identified in early stage LCC tumors could be useful in order to overcome these obstacles.

Therefore, we propose that our defined transcriptional signatures could have a potential utility in predicting the response to immunotherapy in LCC tumors. Considering the relevance of the immune landscape for the potential of a NSCLC patient to respond to immunotherapies, we speculate that patients falling into our newly defined pro-immunogenic group of LCC tumors might be better candidates to receive these treatments, whereas patients classified as pro-tumorigenic might not benefit so much.

Based on this selection of genes, pro-immunogenic and pro-tumorigenic tumors could be clearly identified and showed an opposing transcriptomic profile (Figure 5E). This set of biomarkers may allow an accurate LCC tumor classification to be made and a prediction of response to immunotherapy that could be eventually implemented in clinical practice.

## 4. Discussion

NSCLC management has changed markedly with recent advances in immunotherapy. However, there is still room for improvement and a clear need to accurately identify immunotherapy responders through the definition of better predictive biomarkers. Most research in this area is focused on adenocarcinomas and squamous cell carcinomas, the two main histologies of NSCLC [25,48,49]. However, an exhaustive description of Large Cell Carcinoma, which shows a poor prognosis and a more aggressive behavior, is still lacking [6,9]. An urgent unmet need is for comprehensive studies to be performed on LCC in order to translate the discoveries of predictive biomarkers for response to immunotherapy into clinical benefits for this subset of patients.

In this work, we established a cohort of 18 clinically annotated, early-stage LCC tumors, collected from 2002 to 2011 in our hospital, and described them thoroughly from a multiparametric perspective, taking into account molecular, immune and clinical characteristics. Our main goal was to define novel subgroups of LCC tumors with distinctive molecular, immune and clinical features and to identify specific biomarkers of the cited subgroups that could improve the prediction of response to immunotherapy with checkpoint blockers.

In terms of immune infiltration, our cohort showed high positivity for macrophages (CD68^+^), T CD8^+^ cells and B cells. In contrast, the cases positive for T CD4^+^ cells or PD-L1 were less frequent. With respect to PD-L1 expression, only 17% of tumors were positive, which was clearly below previous descriptions of independent cohorts of resected LCC [8]. PD-L1 evaluation has several associated pitfalls [21,31] that are currently being addressed [32,33]. For example, PD-L1 is expressed by several cell populations and compartments and it can be altered by therapy regimes or tumor heterogeneity, among others. These limitations may explain, at least in part, the differences observed. When evaluating the correlation between PD-L1 expression and the infiltration of immune cells, we observed that all PD-L1 positive tumors (N = 3) were positive for T CD8^+^ infiltration and negative for T CD4^+^ infiltration. However, these observations require further validation in larger LCC cohorts with more PD-L1^+^ tumors.

Taking into consideration the histological markers evaluated, nine tumor samples did not express any adenocarcinoma or squamous cell carcinoma markers and were thus re-classified as LCC-Null according to the 2015 WHO guidelines. Four tumors were positive for TTF-1 expression and five were positive for p40 and were therefore re-classified as LCC-ADC or LCC-SCC, respectively, following the approach of a previously published report [8]. Importantly, FFPE blocks were appropriately managed and all the tumors in our cohort were thoroughly examined to rule out biphasic characteristics, ensuring a sole histology. Beyond a pathological perspective, the frontiers of LCC classification are diffuse and yet to be properly defined as discussed in the literature. For example, the cited report proposed that the LCC-ADC subtype is an entity with different phenotypic and genetic characteristics with respect to classic ADC [8]. It also discussed that the LCC-Null subtype could be considered as a TTF-1 negative adenocarcinoma [8]. Other works suggest that the LCC histology could eventually become extinct with the recent re-classifications of these tumors into poorly differentiated forms of the major subtypes of NSCLC [9]. These key questions are yet to be solved and will likely imply new approaches for the clinical management of these subtypes of patients.

We defined two novel subgroups of LCC tumors based on the expression of genes involved in the tumor–immune system communication and subsequently compared their clinical, molecular and immune characteristics. Importantly, the LCC subtypes (LCC-ADC, LCC-SCC and LCC-Null) did not imply a bias as they were not statistically associated with any of the defined subgroups. Tumors defined as pro-tumorigenic showed a trend to accumulate certain molecular aberrations and were found in a higher proportion of relapsed patients. On the other hand, tumors defined as pro-immunogenic had a higher positivity for all immune markers (except for CD68^+^ macrophages), as evaluated by immunohistochemistry, especially with regards to T CD8^+^ cells, and encompassed all tumors with a high infiltration of T CD4^+^ and B cells. Interestingly, one report showed that a higher infiltration of B cells was associated with a better response to immunotherapy [50]. Of note, the pro-immunogenic subgroup exhibited unique molecular alterations, gathering all tumors with alterations in *ALK*, *ARID1A*, *MET*, *KRAS* or *FGF3*. To this end, there are no targeted therapies directed to all *KRAS* mutations, the most common genetic alteration in NSCLC, but these tumors are considered good candidates for immunotherapies [51].

In agreement with our previous findings, pro-tumorigenic tumors show upregulated pro-tumoral genes involved in processes such as proliferation, adhesion or migration. In contrast, genes with increased expression in pro-immunogenic tumors were associated with immune functions like antigen processing and presentation, immune signaling and stimulation or lymphocyte activation and infiltration, among others. One study reached similar conclusions when analyzing immune hot and immune cold NSCLC tumors characterized by specific molecular, immune and clinical features. Immune hot tumors showed higher T CD8^+^ infiltration and the upregulation of genes related to pro-immune functions such as T cell trafficking and cytotoxicity. Immune cold tumors had lower levels of T CD8^+^ infiltration and lower expression of the cited immune functions [48]. Of note, the cited article reported that certain immune hot-related gene expression signatures were associated with the response to immunotherapy. Nonetheless, recent studies suggest that classifying tumors into hot and cold subsets according to the immune landscape could be a simplistic perspective, and that additional factors should be evaluated [52].

Taken together, these observations suggest that an association exists between the molecular genotype, the immune phenotype and the clinical features of tumors, as previous research reported [48,49,53]. The association of infiltrating immune cells with longer overall survival has been widely demonstrated in cancers with different histological features and anatomical location [42]. When evaluating the possible clinical implications of the LCC pro-immunogenic and pro-tumorigenic subgroups, pro-immunogenic tumors showed a trend towards a better survival, both in terms of PFS and OS, the full extent of which was limited by the cohort size. An additional analysis of PFS, according to the stage of the tumors and their transcriptomic subgroup, suggested these two factors had an impact on relapse.

We identified a transcriptional signature composed of 20 differentially expressed genes between the pro-immunogenic and the pro-tumorigenic LCC tumors that could be used to classify LCC patients into the defined subgroups. Those genes characteristic of the pro-immunogenic tumors were mainly involved in promoting immune responses while those distinctive of pro-tumorigenic tumors favored tumor progression and development.

The ultimate goal of our study was to identify predictive biomarkers of response to immunotherapy in LCC. As previously discussed, these biomarkers should be ideally defined in advanced stage LCC patients treated with immunotherapy. However, the amount of LCC tumors following these treatment regimens that could undergo a comprehensive description successfully is scarce. In this context, thorough descriptions of early stage LCC samples could provide a solid foundation to extrapolate findings into the advanced disease setting.

Therefore, we propose that our defined transcriptional signatures could potentially improve the prediction of response to immunotherapy in LCC. It could be hypothesized that pro-immunogenic tumors are better candidates to receive immunotherapy, as from both a transcriptomic and immune infiltrate perspective they seem more prone to stimulate immune responses. To this end, it has been demonstrated that a previous ongoing immune response is crucial in order to respond to immunotherapy [42,52].

The potential clinical implications of some of the genes of our transcriptional signatures have already been suggested in previous studies. For example, an elevated expression of *FCRLA* in lung adenocarcinoma, among other genes, was associated with a better response to immunotherapy [50]. Expression of *CD137 (TNFRSF9)* and *PSBM9* in NSCLC tumors was increased in immunotherapy responders [49], while an elevated expression of *PSBM9* was associated with higher PFS. Interestingly, *PSBM9* was not selected under our criteria but almost reached statistical significance in our study (FDR = 0.01 and log_2_ Fold Change = 0.96, upregulated in the pro-immunogenic subgroup) which suggests that our biomarker filtering criteria could perhaps be reassessed in order to identify additional genes with potential as biomarkers. In contrast, previous research proposed that a highly or poorly proliferative tumor microenvironment was associated with a worse response to immunotherapy in NSCLC compared to moderately proliferative tumors. This proliferative status was assessed by the evaluation of expression of a 10-gene signature, including some of our pro-tumorigenic biomarkers such as *CCNB2*, *KIAA0101*, *MAD2L1* and *TOP2A* [54]. Of note, *MKI67*, a proliferation marker, was also significantly upregulated in the pro-tumorigenic tumors of our study, supporting the proliferative-prone status of this subset. The association of our selected markers and potential responses to immunotherapy is not exclusive of NSCLC. It was described, for example, that melanoma responders to antiPD-1 therapy had significantly increased numbers of CD69^+^ NK cells [55]. Another study reported that *CRTAM* was the most upregulated gene in responders to immunotherapy with diffuse large B-cell lymphoma [56].

Precision oncology is defined as the designation of the right treatment for the right patient. In the context of NSCLC, great efforts are ongoing to provide more accurate and individualized descriptions of adenocarcinomas and squamous cell carcinomas as these are the main pathology in approximately 90% of NSCLC patients. However, current approaches might be omitting less frequent histologies such as LCC. Our study reports that comprehensively describing tumors from LCC patients can result in the identification of key molecular and immune characteristics that could be exploited to increase the benefit obtained from immunotherapy. Nevertheless, more robust data will require similar research to be performed in larger patient cohorts. Furthermore, the predictive value of the selected biomarkers needs to be validated in patients who have previously received immunotherapy. Previously mentioned research has demonstrated the remarkable value of multiparametric studies to improve immunotherapy outcomes. It must be ensured that all tumor types, and ultimately all patients, benefit from these new research approaches that are paving the way towards personalized medicine.

## Figures and Tables

**Figure 1 jcm-11-01500-f001:**
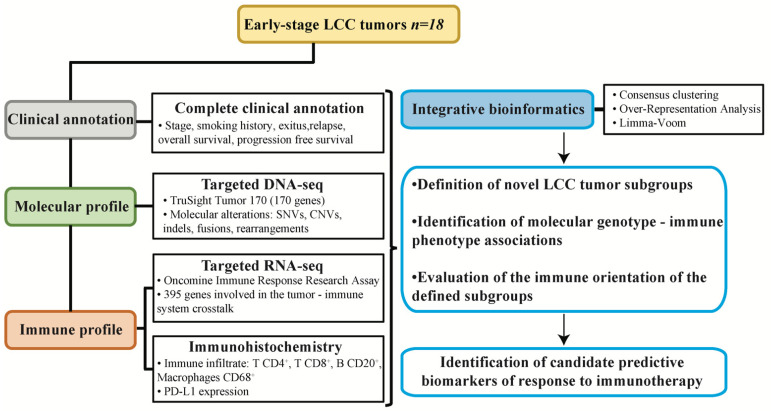
Experimental design of the study. Samples from resected early-stage large cell carcinoma (LCC) of a clinically annotated cohort of 18 patients were subjected to molecular and immune profiling and integrative bioinformatics to find associations permitting patient subgroup classification according to potential responses to immunotherapy.

**Figure 2 jcm-11-01500-f002:**
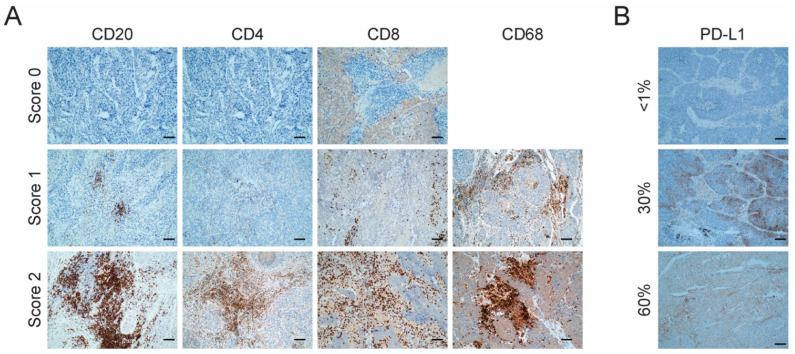
Characterization of tumor immune infiltrate and PD-L1 expression. (**A**) Representative images of the expression of immune markers for CD20, CD4, CD68 and CD8 by immunohistochemistry in LCC tumors. Score 0: <1%, negative; Score 1: 1–10%, low expression; Score 2: >10%, high expression. Scale bar = 2 µm. No sample had a Score of 0 for CD68; (**B**) Representative images of the expression of PD-L1 by immunohistochemistry in LCC tumors. Staining ≥1% was considered to represent positivity. Scale bar = 2 µm.

**Figure 3 jcm-11-01500-f003:**
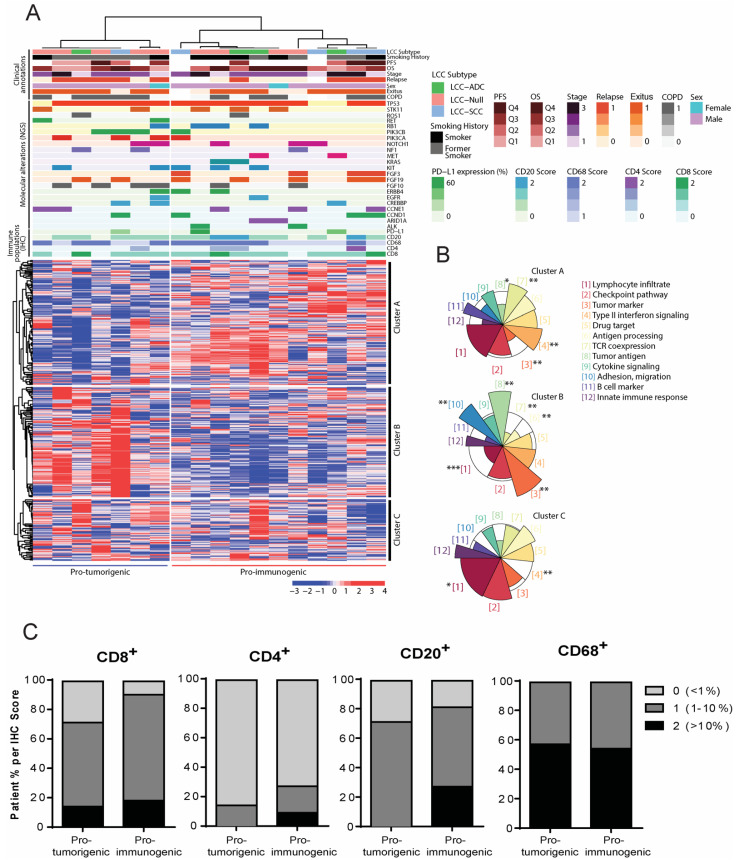
Definition of novel subgroups of Large Cell Carcinoma (LCC) tumors based on molecular and immune profiles. Large Cell Carcinomas (N = 18). (**A**) Heatmap of expression of genes involved in tumor–immune system communication. Groups of tumors are shown in the horizontal axis and clusters of genes in the vertical axis as defined by consensus clustering. Molecular, immune and clinical annotations are shown above the heatmap (see legend). For gene annotations, molecular alterations are represented as colored sections. The heatmap scale is expressed in percentiles; (**B**) Over-Representation Analysis (ORA) bringing together onco-immune functions. Each colored sector represents a functional category whose size illustrates over- or under-representation of genes in that cluster within a specific function. Over-represented functions allow patient groups to be classified as pro-tumorigenic or pro-immunogenic according to their tumor immunogenicity. The over or under-representation of each functional category was statistically evaluated with a Fisher’s exact test. * (*p* < 0.05), ** (*p* < 0.01), *** (*p* < 0.001); (**C**) Immune cell population distribution in the pro-immunogenic and pro-tumorigenic LCC subgroups.

**Figure 4 jcm-11-01500-f004:**
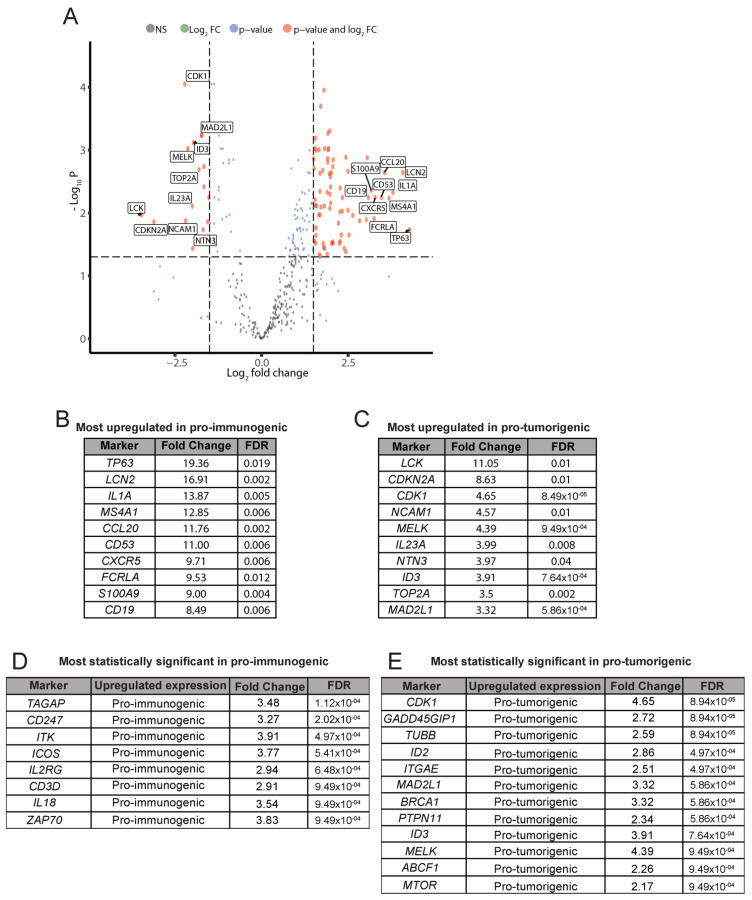
Analysis of differential gene expression. Pro-immunogenic vs. Pro-tumorigenic subgroups. (**A**) Volcano plot of differentially expressed genes between the pro-immunogenic group and the pro-tumorigenic group of LCC tumors. A false discovery rate (FDR) ≤ 0.05 and log_2_ Fold Change ≥ |1| were required to reach statistical significance; (**B**) Selection of the most upregulated genes in the pro-immunogenic tumors; (**C**) Selection of the most upregulated genes in the pro-tumorigenic tumors; (**D**) Selection of the most statistically significant upregulated genes in the pro-immunogenic tumors; (**E**) Selection of the most statistically significant upregulated genes in the pro-tumorigenic tumors.

**Figure 5 jcm-11-01500-f005:**
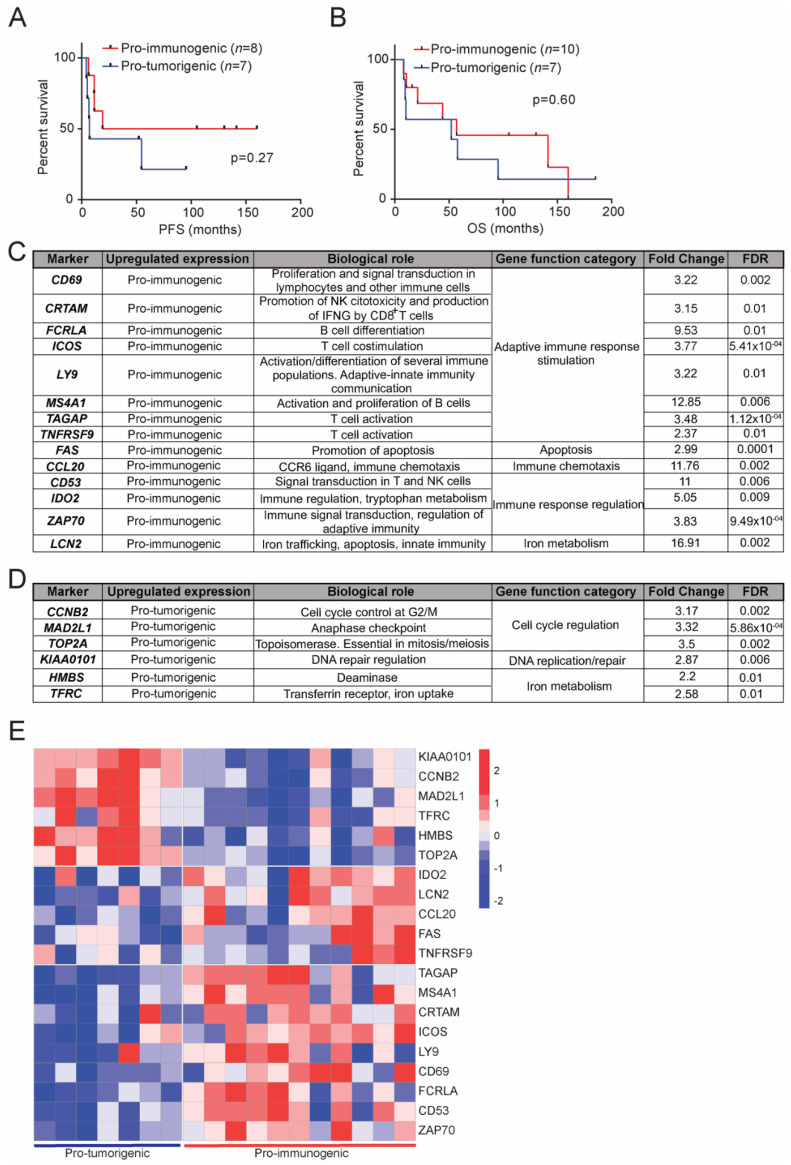
Survival analysis of the defined LCC subgroups and selection of candidate predictive biomarkers for response to immunotherapy. Kaplan–Meier analysis of progression free survival (PFS) (**A**); and overall survival (OS) (**B**) between pro-immunogenic (red) and pro-tumorigenic (blue) tumors. OS and PFS data could not be retrieved from all 18 LCC patients. Statistical analyses were performed with a log-rank (Mantel–Cox) test for which a p-value below 0.05 was considered significant; (**C**) Selected candidate predictive biomarkers of response to immunotherapy characteristic of pro-immunogenic LCC tumors. FDR = False Discovery Rate; (**D**) Selected candidate predictive biomarkers of response to immunotherapy characteristic of pro-tumorigenic LCC tumors; (**E**) Heatmap of expression of the selected biomarkers, grouped by clusters, in pro-tumorigenic tumors (left) and pro-immunogenic tumors (right).

**Table 1 jcm-11-01500-t001:** Demographic and clinical characteristics of the LCC cohort (N = 18).

Characteristic	N	(%)	Characteristic	N	(%)
Total LCC cases	18	100.00	Total LCC cases	18	100.00
Sex			Stage		
Male	16	88.89	I	5	27.78
Female	2	11.11	II III	9 4	50.00 22.22
Average age at diagnosis	65.7	NA	Neoadjuvant therapy		
Smoking History			Yes	1	5.56
Current smoker	7	38.89	No ND	15 2	83.33 11.11
Former smoker	9	50.00	Adjuvant therapy		
Never smoker	0	0.00	Yes	9	50.00
ND	2	11.11	No ND	7 2	38.89 11.11
Pack-year group			Exitus		
<10	0	0.00	Yes	13	72.22
10 to 20	1	5.56	No	4	22.22
20 to 40	1	5.56	ND	1	5.56
>40	5	27.78	Average OS (months)	65.4	NA
NA	0	0.00	Relapse		
ND	11	61.11	Yes	9	50.00
COPD			No ND	6 3	33.33 16.67
Yes	10	55.56	Average PFS (months)	54	NA
No	6	33.33			
ND	2	11.11			

ND stands for “Not Determined”. NA stands for “Not Available”.

**Table 2 jcm-11-01500-t002:** Immunohistochemistry scoring for immune markers in both LCC defined subgroups.

Marker	Pro-Immunogenic Tumors N = 11						Pro-Tumorigenic Tumors N = 7					
Score	0	1	2	Positive (%)	Negative (%)	Avg. Score	0	1	2	Positive (%)	Negative (%)	Avg. Score
CD8	1	8	2	10 (90.91)	1 (9.09)	1.09	2	4	1	5 (71.43)	2 (28.57)	0.85
CD4	8	2	1	3 (27.27)	8 (72.73)	0.36	6	1	0	1 (14.29)	6 (85.71)	0.14
CD68	0	5	6	11 (100)	0 (0)	1.55	0	3	4	7 (100)	0 (0)	1.57
CD20	2	6	3	9 (81.82)	2 (18.18)	1.09	2	5	0	5 (71.43)	2 (28.57)	0.71
PD-L1	NA	NA	NA	2 (18.18)	9 (81.82)	NA	NA	NA	NA	1 (14.29)	6 (85.71)	NA

Cases with more than 1% of positive staining were considered to be positive. Avg. Score is the average score value in each subgroup for each marker. Score 0 < 1%, Score 1 ≥ 1%; <10%, Score 2 ≥ 10%. NA stands for “Not Available”.

## Data Availability

The data presented in this study are available in the article or uploaded as Appendix A.

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
