# Peer review of "Comprehensive Characterization of Human Lung Large Cell Carcinoma Identifies Transcriptomic Signatures with Potential Implications in Response to Immunotherapy"

_jcm, 2022, doi:10.3390/jcm11061500_

Round 1
Reviewer 1 Report
Large cell carcinoma (LCC) is a subtype of NSCLC, representing around 10% of NSCLC cases. Diagnosis of LCC requires examination of surgically resected tumor, limiting the understanding of LCC molecular subtypes and micro-environment characteristics. In this manuscript, the authors obtained 18 surgically resected LCC samples, with most samples being treatment-naïve. Mutational and transcriptional phenotypes were profiled using FFPE samples, along with Immunohistochemistry staining of representative immune cell types. The authors classified the 18 LCC cases into pro-immunogenic and pro-tumorigenic subtypes, with extensions into PFS and OS of different subtypes. Overall, the manuscript provides a resource to understand the genetic, transcriptomic, and microenvironment features of LCC. Considerations to potentially improve the article are listed as follow:
- The authors may consider citing other articles that have performed sequencing analysis on LCC samples. A summary of previous work and their findings will contribute to understanding the background and questions of the field.
- Figure qualities for IHC results in fig 2 are very poor.
- Since PD-L1 status may indicate pre-existing immunity, the correlation between PD-L1 score and immune infiltration (especially CD8 T cells) should be examined. Related to figure 2.
- For figure 3B, representative genes for cluster A, B and C, especially genes with immune functions or indicative of tumor fraction should be listed or plotted separately, in addition to the current gene sets. Although differentially expressed genes are listed in figure 4, it is still necessary to list representative genes in figure 3.
- Section 3.5 is confusing. Biomarkers for immunotherapy are best derived from cohorts undergoing immunotherapy, but in the current manuscript the majority of patients are treatment naïve. Indeed, LCC can be relatively rare, with LCC patients undergoing immunotherapy very limited. But in the current manuscript, the authors compared PFS and OS for treatment-naïve patients only, between pro-tumor/pro-immunogenic groups. No significant difference was observed between the two groups, and thus the extension into predictive biomarkers for immunotherapy lacks a solid foundation. Finally, the article is entitled “Identification of candidate biomarkers for predicting response to immunotherapy in human lung Large Cell Carcinoma”, but the bulk of its contents are to depict the genetic and transcriptomic features of treatment-naïve samples, with only the last section trying to establish connection with immunotherapy. The “identification of biomarker…” part of the title can be misleading, and should be adjusted.
Reviewer 2 Report
This manuscript may be a beneficial examination for exact medical developments of the immune checkpoint inhibitor of future lung large cell carcinoma. Indeed, I am interested in immuno-molecular profile of resected LCC at surgical time. However, a problem is still left unfinished about the evaluation with the treatment response.
It is useful and I think that it was able to be a valuable report by major revision. Here is a summary of steps that I would recommend:
Major revision:
1. P13 L438 I am interesting in PFS curve. This is early stage LCC cohort, but I think that it is necessary for the distribution of the stage of each group to describe it in manuscript simply. Indeed, figure 3A shows stage of each groups, it is too small.
2. In table 1, are mutations detected by commercial exmanation ? I think that driver mutations (EGFR, KRAS, ROS-1, ALK) are detected by panel test at figure 1. Clinician may feel that the prevalence of these driver mutations in the gene might be too high. I think these mutation may contains meaningless minor mutation in Table S2. Therefore, you need add that these mutations may lead treatment adaptation at P6 L297.
Minor revision:
1. P7 Fig.1 Please rivise "Lima-Voom" for "Limma-Voom". If you can, replace more high resolution figure, please.
2. P14 Fig 5 Please rivise "p=0,27" for "p=0.27" and "p=0,60" for "p=0.60".
Round 2
Reviewer 2 Report
This manuscript is a beneficial examination for exact medical developments of the immune checkpoint inhibitor of future lung large cell carcinoma. Thank you for having you revise it carefully.
I feel that the report became more attractive, in particular that "Figure S3D" is very interesting.
It became easy to see table 1 for a clinician very much, too.
Minor revision:
1. Fig. S3A Please add Stage III line.
2. Fig. S3C Please revise each line colors.
